# Trajectory of Psychosocial Measures Amongst Informal Caregivers: Case-Controlled Study of 1375 Informal Caregivers from the English Longitudinal Study of Ageing

**DOI:** 10.3390/geriatrics5020026

**Published:** 2020-04-27

**Authors:** Toby Smith, Amanda Saunders, Jay Heard

**Affiliations:** 1Queen’s Building, Faculty of Medicine and Health Sciences, University of East Anglia, Norwich NR4 7TJ, UK; 2Nuffield Department of Orthopaedics, Rheumatology and Musculoskeletal Sciences, University of Oxford, Oxford OX3 7LD, UK; 3Physiotherapy Department, Addenbrookes Hospital, Cambridge University Hospitals NHS Foundation Trust, Cambridge SE13 6LH, UK; amanda.saunders@addenbrookes.nhs.uk; 4Physiotherapy Department, Lewisham Hospital, Lewisham and Greenwich NHS Trust, London CB2 0QQ, UK; jay.heard@nhs.net

**Keywords:** caregiver, family support, community independence, older people, trajectory

## Abstract

Informal caregivers provide vital support for older adults living in the community with chronic illnesses. The purpose of this study was to assess the psychosocial status of informal caregivers of community-dwelling adults over an eight-year period. Informal caregivers of adult care-recipients were identified from Wave 1 of the English Longitudinal Study of Ageing (ELSA) cohort. Multivariate regression analysis models were constructed to assess the association between participant’s psychosocial characteristics and informal caregiving. Multilevel modelling explored the psychosocial changes between caregivers and non-caregivers over eight years. 1375 informal caregivers and 2750 age-matched non-caregivers were analyzed. Self-reported loneliness (Odd Ratio (OR): 0.26; 95% confidence intervals (CI): 0.01–0.51) and relationship status (OR: 0.36; 95% CI: 0.16–0.46) were independently associated with caregiving. Caregivers were more socially isolated with less holidaying abroad (OR: 0.51; 95% CI: 0.35–0.66), attendance to church (OR: 0.30; 95% CI: 0.11–0.49), or charity groups (OR: 0.35; 95% CI: 0.14–0.55). On multilevel analysis, over time (eight-years), caregivers reported greater loneliness (*p* < 0.01), change in relationship status (*p* = 0.01) and reduced control, autonomy, and pleasure (*p* ≤ 0.01) compared to non-caregivers. Given the deleterious effects caregiving can place on health and wellbeing, further interventions are required to improve these psychosocial factors.

## 1. Introduction

Informal caregivers provide vital, unpaid support to maintain independent living for older people living in the community [1]. Caregivers have been defined as ‘carers, who may or may not be family members, are lay people in a close supportive role who share in the illness experience of the patient’ [2]. They may provide an array of different roles of support, from assistance with activities of daily living such as washing, dressing, bed–chair transfers, cooking, and feeding, or more complex tasks such as finances, correspondence, and shopping [3]. They may also be expected to provide emotional support [2]. This group of individuals are therefore heterogeneous both in their relationships to caregivers, being family members or friends [3], in their characteristics both in age and employment status and other life commitments, but also in the roles and tasks which these individuals provide care-recipients [4,5]. Caregivers are expected to support their family members or friends more due to an increasing shift from professional to informal care [6].

Previous literature has indicated that informal caregiving is associated with poorer psychological wellbeing and reduced perceived social worth and loneliness [7,8]. The latter is particularly important given that loneliness can negatively influence higher-order cognitive processes such as attention, memory, emotional regulation, and logical reasoning [9]. Loneliness and social isolation can present as depression, boredom, or self-deprecation, along with increased risk of dementia, particularly amongst older caregivers [10,11,12]. Both loneliness and social isolation have been associated with increased frequency of older adults’ visits to their doctor [12]. Burden and consequences on older caregivers with health conditions may be particularly important given they frequently present with poor general health through physical disability and cognitive impairment [13,14]. Such health challenges extenuate the difficulties a caregiving dyad may face in maintaining independence and their desired quality of life [13,14].

Various sociological models have explained caregiver/care-recipient lived experiences. These include: the Social Ecological Theory [15], where caregiving is influenced by various social contexts; the Life Course Theory [16], where caregiving has discrete entry, exit, and transition points dependent on time; and the Pearlin Stress Process Model [17], which acknowledges that caregivers experience, appraise, and cope with care demands through moderators to develop a positive or negative caregiving experience. Engel’s [18] biopsychosocial model of health encapsulates numerous elements of these models, where the interconnections of biology, psychology, and socio-environmental factors can be used to understand the dynamic construct caregiving has on the caregiver, care-recipient, health and social care systems, and wider society. Given these contextual factors, this is a valuable model when investigating the caregiving dyad. However, there remains limited evidence how these change over time [19,20].

The purpose of this analysis was to investigate the trajectories of psychosocial outcomes for informal caregivers in England. The findings of this will be valuable to better understand what psychosocial features are important over time for these individuals, and whether interventions are needed for the health and wellbeing of informal caregivers. Supporting the caregiver needs more effectively, with strong a caregiver-care-recipient dyad, has importance in both promoting the independence of the older people from formal health services, and to reduce economic and social burden on national health services to support both formal care and more costly acute care during periods of exacerbation.

## 2. Materials and Methods

The Strengthening the Reporting of Observational Studies in Epidemiology (STROBE) guidelines were followed in the reporting of this comparative prospective cohort study [21].

### 2.1. Cohort

Data were drawn from the English Longitudinal Study of Ageing (ELSA) cohort. ELSA is an ongoing, national cohort study of community-dwelling adults born on or before 29th February 1952. It is a nationally representative sample of the community-dwelling population living in England, aged 50 years or older on enrolment [22]. ELSA aims to examine the relationship between health with economic activity, social participation, physical activity and lifestyle behaviors, productivity, networks, and sport [23]. From the 2002/2003 inception, participants have been followed-up every two years.

Ethical approval was gained from the London Multi-Centre Research Ethics Service (Reference number: MREC/01/2/91). Anonymized unlinked data for this study was provided by the UK Data Service (https://www.ukdataservice.ac.uk).

### 2.2. Participant Identification

Participants were identified as informal caregivers from ELSA Wave 1 if they self-reported that they cared for/supported a care-recipient for functional, Activities of Daily Living (ADL; e.g., walking, feeding, dressing, toileting, bathing, and transfers), or Instrumental Activities of Daily Living (IADLs; e.g., managing finances, transportation, shopping, preparing meals, household chores and maintenance, managing medications, and correspondence). Participants who were caregivers for only children were excluded from the analysis. Caregiving status was ascertained across data collection waves (Waves 1 (2002/2003) to Wave 5 (2010/2011)) to ensure participants were caregivers across each time-point.

A non-informal caregiver cohort was gathered from the Wave 1 ELSA cohort. These were age-matched to the informal caregiver cohort by a ratio of 2:1. Only caregivers or non-caregivers were included if a full-data set was available for the outcomes of interest.

### 2.3. Data Identification

Demographic characteristics for caregivers were gathered, including age, gender, ethnic classification (white/non-white), relationship status, and occupational status. We identified the relationship of the caregiver to care-recipient and the number of hours caregiving provided in the previous week.

Psychosocial features were gathered given their previously reported association to informal caregiving [24,25]. Social measured included participant’s social and cultural attendance (cinema, eating out, art gallery/museum attendance, theatre, opera or concert attendance), work status, holidaying, use of the internet and emailing, and attendance/membership of local sporting, religious, political, charitable, or educational groups. We also assess the number of people who lived within the caregiver’s household. Psychological measures included self-reported depression, self-reported loneliness, and the General Health Questionnaire-12 (GHQ-12) [26], which was used to assess mental well-being (range 0–36; higher scores indicating worse condition). There were data available to assess CASP-19 [27] from Waves 2 to 5 (range 0–57; higher scores indicating greater satisfaction with quality of life). This is a quality of life scale for use in older adults and assesses the domains of control, autonomy, pleasure, and self-realization [27].

### 2.4. Data Analysis

Variables were descriptively analyzed through mean and standard deviation (SD) values for continuous data, and frequency and percentages for categorical responses, stratified by caregiving status.

Univariate logistic regression analyses were performed on all variables. Being a caregiver was the dependent variable. Variables that reached a statistical significance of *p* < 0.20 on univariate analysis were brought-forward to multivariate analysis. The construction of the multivariate analysis models were based on the biopsychosocial model [18]. Three cumulative regression models were constructed: Model 1 included demographic/biological–physical health factors; Model 2 added psychological factors; and Model 3 added social factors. Data were presented as odds ratios (OR), 95% confidence intervals (CI) and *p*-values. Statistical significance was deemed where *p* < 0.05.

Multilevel modelling determined whether the ‘time’ variable (levels = Wave 1 to 5) was significant between caregivers and non-caregivers. The model was built by including all the variables reported as independently associated with caregiving on Model 3 of the multivariate analysis (self-reported loneliness, relationship status, cinema attendance, holiday abroad, church membership, charity group membership). There were insufficient data to perform the trajectory analysis on GHQ-12 data, therefore perceived strain not assessed. However, the CASP score was assessed from Wave 2 to 5 for total score, control CASP, autonomy CASP, pleasure CASP, and self-realization CASP. Self-realization CASP was excluded from the final multi-level model due to collinearity. All analyses were undertaken using Stata Statistical Software, Release Version 16.0 (StataCorp LP, College Station, TX, USA).

## 3. Results

In total, 1375 informal caregivers and 2750 age-matched non-caregivers were analyzed. Figure 1 illustrates how the cohort was derived.

### 3.1. Characteristics of Informal Caregivers vs. Non-Informal Caregivers

Table 1 illustrates the results of the psychosocial univariate analysis. Demographic factors associated with caregiving included ethnicity (*p* < 0.01), gender (*p* < 0.01), relationship status (*p* < 0.01), numbers of people living within the respondent’s household (*p* < 0.01), self-reported health (*p* = 0.03), self-reported chronic diseases (*p* = 0.10), being often ‘troubled by pain’ (*p* < 0.01), and cognitive measures including immediate word recall (*p* = 0.05), fluency (*p* = 0.02), numeracy (*p* = 0.02), and self-reported loneliness (*p* = 0.06).

Sociological factors associated with caregiving were cinema attendance (*p* < 0.01), eating out (*p* < 0.01), visiting an art gallery/museum (*p* = 0.03) or theatre (*p* = 0.01), holidaying abroad (*p* < 0.01), going on daytrips (*p* = 0.02), using the internet or emailing (*p* = 0.02), and being a member of a residential group (*p* < 0.01), church or religious group (*p* < 0.01), charitable organization (*p* < 0.01) or education/arts/music class/group (*p* = 0.01).

Psychological factors measured using the GHQ-12 associated with caregiving included concentration (*p* < 0.01), loss of sleep (*p* < 0.01), perceived strain (*p* < 0.01), inability to overcome difficulties (*p* < 0.01), ability to enjoy life (*p* = 0.01), problem-solving ability (*p* < 0.01), feeling unhappy or depressed (*p* < 0.01), losing self-confidence (*p* < 0.01), and perceived happiness (*p* = 0.01).

Table 2 demonstrates the results of the multivariate analysis. Model 3 reports the combined psychosocial analysis. From this, people who were non-white were less likely to be caregivers (OR: 1.28; 95% CI: 1.20–1.37), males were 75% less likely to be caregivers (OR: 0.25; 95% CI: 0.09–0.41), caregivers were 74% less likely to report loneliness (OR: 0.26; 95% CI: 0.01–0.51), and 64% less likely to be single (OR: 0.36; 95% CI: 0.16–0.46). Caregivers were also 49% less likely to have been holidaying abroad in the last 12 months (OR: 0.51; 95% CI: 0.35–0.66).

Caregivers were more likely to attend church groups (OR: 0.30; 95% CI: 0.11–0.49) or charity organizations (OR: 0.35; 95% CI: 0.14–0.55). Caregivers were 77% less likely to report strain compared to non-caregivers (OR: 0.23; 95% CI: 0.09–0.37). All other variables were reported not to be independently associated with informal caregiving.

### 3.2. Trajectory Analysis

As Figure 1 illustrates, it was possible to analyze the trajectories of 777 caregivers and 1463 non-caregivers for psychosocial variables identified as independently associated with caregiving from the multivariate analysis and CASP measures. The results of these are summarized in Table 3.

Whilst there was no difference in the eight-year trajectories for holidaying abroad, church membership, charity organization group membership, and total CASP score between the caregiver and non-caregiver groups, there were differences between the groups in the trajectories for the remaining five variables. Whilst the multivariate analysis suggested caregivers were less lonely compared to non-caregivers, this reversed over time, where caregivers more frequently reported loneliness (Figure 2). Relationship status was significantly different between the groups over time. Caregivers were more frequently married or co-habiting at Wave 1 but less likely by Wave 5 (Wave 1: 82.2% vs. 74.5%; Wave 5: 66.4% vs. 70.4%; *p* = 0.01; Figure 3).

There were significant differences between the trajectories of caregivers and non-caregivers for control, autonomy, and pleasure CASP domains. Figure 4 illustrates the significant difference (*p* < 0.01) between the two groups more notably for Waves 2 and 3 (Wave 2: 8.2 vs. 8.5; Wave 3: 7.6 vs. 8.0). Figure 5 illustrates the difference in autonomy CASP scores between the caregiver groups (*p* < 0.01). Whilst CASP pleasure remained the same throughout the Wave 2 to 5 for the caregiver group (13.3), it declined in the non-caregiver group (Figure 6). Although these were statistically significant (*p* = 0.01), there was no clinically meaningful difference. Finally, CASP self-realization was not included in the multilevel model due to collinearity. However, Figure 7 illustrates the difference where non-caregivers reported greater scores than caregivers for Waves 2, 3, and 4 (Table 3).

There were no differences in basic demographic characteristics for caregiver or non-caregiver cohorts between the cross-sectional to trajectory analyses (Table 4). This indicates a low risk of selection bias in the trajectory analyses from the overall cohort.

## 4. Discussion

The findings of this study indicate differences in the perceived psychosocial status of caregivers compared to age-matched non-caregivers. Ethnicity, gender, and being married or co-habiting with individuals were all independently associated with caregiving. Similarly, reduced holidaying abroad, but decreased perceived strain were associated with caregiving. Membership of church or charity groups was associated with caregiving. However, over the eight-year follow-up period, caregivers more frequently reported loneliness, lower proportion of married/cohabited relationships, and statistical differences in CASP control, autonomy, and pleasure domains, although these were not clinically significant. The results indicate that interventions to address these psychosocial differences are warranted given their known relationship to poor health and wellbeing status over time.

Whilst not being clinically significant, the CASP pleasure domain demonstrated a decline reported by non-caregivers but maintained static for caregivers over time. This may seem surprising, where caregiving is often perceived as a stressful not pleasurable activity [7,8]. However previous qualitative research has reported the positive experiences that caregiving can offer in some instances [28,29,30]. Where caregiver bonds (often reported through marital happiness but not exclusively) are strong, the act of caregiving may bring a dyad personally closer to one another, offering pleasure and identity to a relationship. The ELSA cohort, whilst being nationally representative, is a self-selecting cohort of individuals who consented to report data to a national cohort study. Whether the proportion of individuals from this cohort reported greater marital or relationship happiness in their caregiving dyad, and if this is typical of the general population, remains unclear.

As acknowledged, the cross-sectional analysis indicated that caregivers reported lower strain compared to non-caregivers. However, this may be a function of the sample selected. Individuals were asked to self-identify as caregivers. As a result, they may have emotionally and practically adapted to this, being in lower perceived ‘strain’ compared to those who provide care, but do not self-identify as such. We are unable to ascertain the ‘perceptions’ towards caregiving activities, of duration and role adaption which may help understand this. However, it raises the question as to whether there are differences in caregiver lived experiences based on the perception of being an experienced or inexperienced caregiver.

There was an independent association between being a member of a church or charity organization and caregiving. Gopalan et al. [31] previously reported the association between caregiving and altruistic characterizes and traits. Whilst membership to these organizations may help to minimize social isolation for caregivers [32,33], it may not necessarily reduce feelings of loneliness, as caregivers in this study were more likely to report perceived loneliness over time. Courtin and Knapp [34] highlighted the importance of distinguishing between social isolation, which is an objective reduction in social relationships, and loneliness, which is the perception of the lack of quality social relationships. This poses a conundrum for determining the most appropriate support for these individuals. Strategies to increase social inclusion may not necessarily address feelings of loneliness if an individual perceives that they are lacking quality, meaningful relationships, although it may provide more opportunities for such relationships to develop [35]. Therefore, it may be important to consider strategies to ensure that caregivers maintain the quality of relationships already present within their social networks, particularly given that loneliness appears to change over time amongst caregivers.

There are two clear clinical applications to these findings. Firstly, the results highlight the detrimental health effects that caregiving may have on psychosocial wellbeing. The results highlight the need to support these individuals to improve resilience and skills which may address the negative consequences of caregiving. Healthier caregivers may provide better caregiving environments to have improved health outcomes for care-recipients. This model requires further investigation. Secondly, the data indicates that those detrimental effects continue over time. Whilst caregiving has been reported as temporal, fluctuating dependent on the dyad and social context, for some identified factors, there remains a deficit. Finally, the results have highlighted a difference in caregiving activities dependent on gender, ethnicity, relationship status, and social engagement. Targeting these individuals for caregiving interventions would be appropriate given these findings.

The strength of this study is the longitudinally collected, nationally representative data. Previous studies have analyzed cross-sectional data [1,3,5]. This longitudinal assessment provides unique insights that there remains a difference that increases between caregivers and non-caregivers for a number of psychosocial variables. Furthermore, caregivers were not selected based on a specific illness or medical condition of the care-recipients. Previous evidence has frequently focused on examining informal caregiver outcomes for caregivers with specific diseases [8,11,19]. Accordingly, this analysis provides new insights to the wider community. However, there remains limitations which should be considered. Firstly, the ELSA cohort provides limited information on the care-recipient. Understanding the caregiving demand on physical or psychological, social, or a mixed support requirement is critical. This factor is important given that previous authors have highlighted greater strain and burden reported by caregivers when caring for people with cognitive impairment compared to people with less unpredictable behavioral challenges [36,37]. Secondly, data were not available to analyze a number of variables which may have important contextual value, most notably whether participants lived in urban or rural communities. Thirdly, there remains limited indication on caregiver burden or the impact of family support. Given that caregiving dyad models have stressed the importance of the dyad on society (Social Ecological Model [15]), which may fluctuate over time (Life Course Model [16]) dependent on the care-recipient’s needs, caregiver capabilities, and health and social care environment (Pearlin’s Stress Process Model [17]), consideration of these with further analyses would be value to explore how these variables interact with the caregiving scenario. Nonetheless, the novel design of this longitudinal study begins generating answers in this field of enquiry.

## 5. Conclusions

There are important differences in the perceived psychosocial status of caregivers compared to age-matched non-caregivers. For a number of psychosocial factors, these remained different between caregivers and non-caregivers over eight years, most notably for greater perceived loneliness. Given the deleterious effects this can have on health, further interventions are required to improve these psychosocial factors. Through a personalized approach, the caregiver/care-recipient dyad may gain health and wellbeing benefits to have a positive benefit for a growing population in the community. Given the recent COVID-19 pandemic and international social distancing/self-isolation policies [38], there is urgent need to implement caregiver interventions focusing on the reported psychosocial challenges.

## Figures and Tables

**Figure 1 geriatrics-05-00026-f001:**
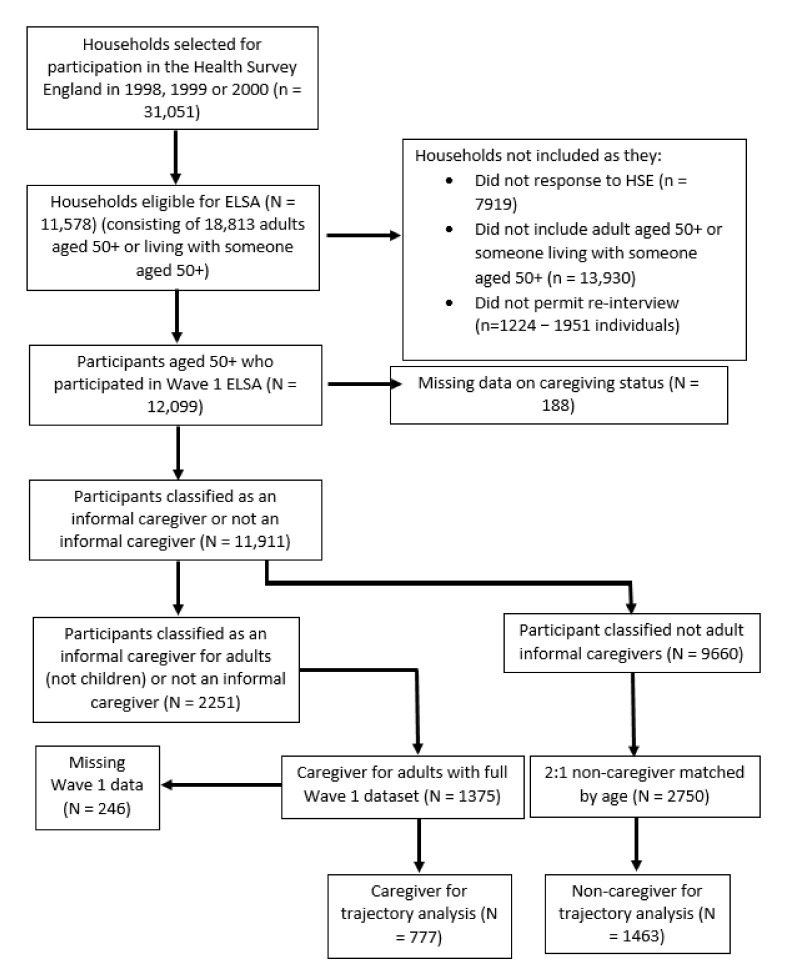
Cohort flow chart of English Longitudinal Study of Ageing (ELSA) Wave 1 participants analyzed as informal caregivers and non-informal caregivers.

**Figure 2 geriatrics-05-00026-f002:**
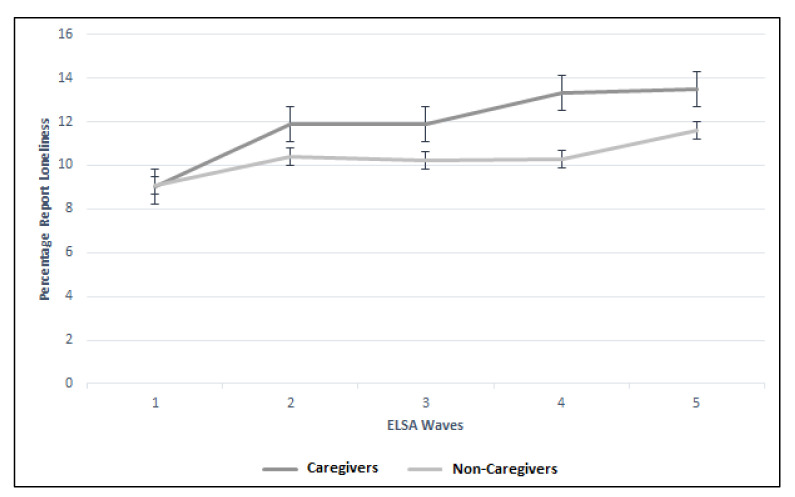
Trajectory of self-reported loneliness between caregiver and non-caregiver cohorts across the five ELSA waves.

**Figure 3 geriatrics-05-00026-f003:**
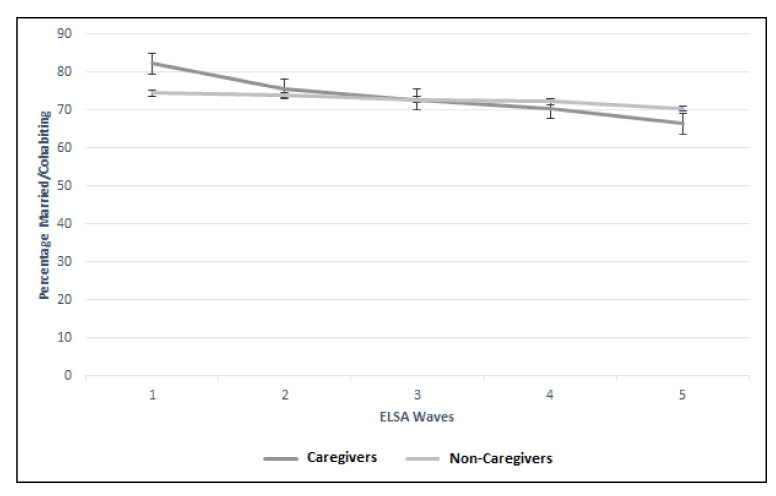
Trajectory of relationship status (married/cohabiting) between caregiver and non-caregiver cohorts across the five ELSA waves.

**Figure 4 geriatrics-05-00026-f004:**
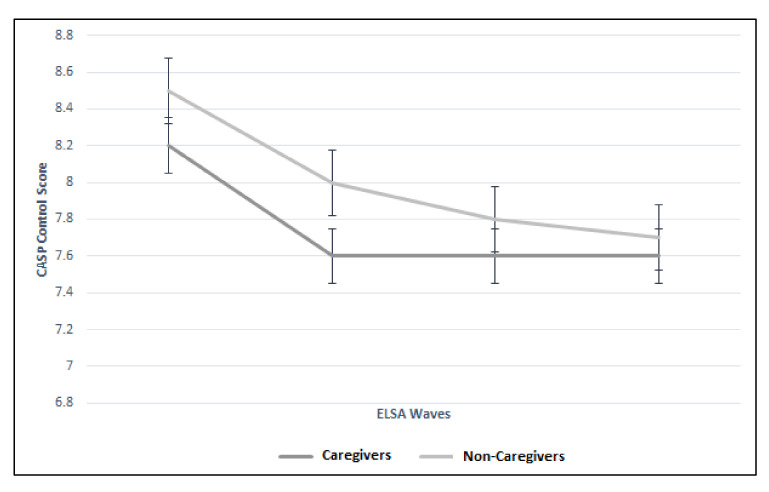
Trajectory of control CASP score between caregiver and non-caregiver cohorts across four ELSA waves.

**Figure 5 geriatrics-05-00026-f005:**
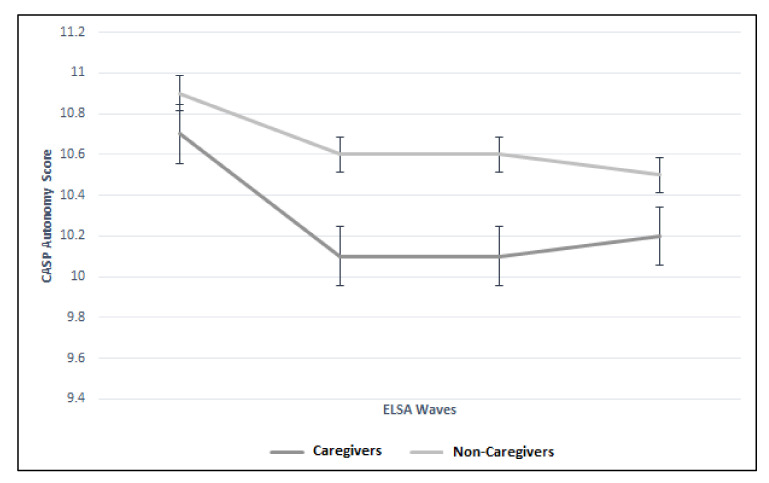
Trajectory of autonomy CASP score between caregiver and non-caregiver cohorts across four ELSA waves.

**Figure 6 geriatrics-05-00026-f006:**
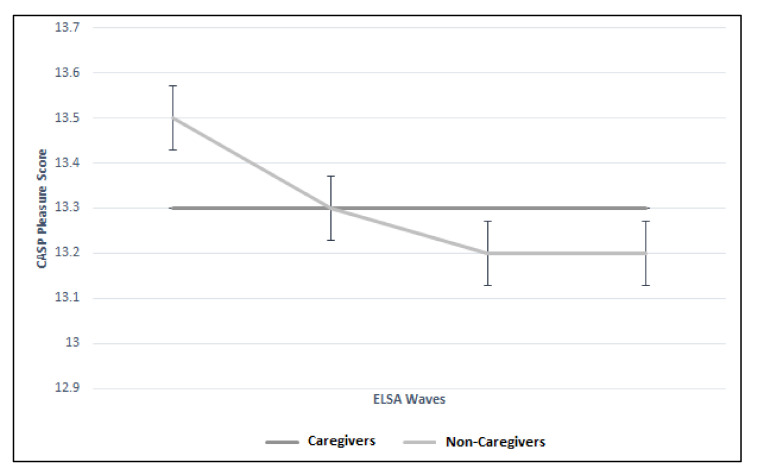
Trajectory of pleasure CASP score between caregiver and non-caregiver cohorts across four ELSA waves.

**Figure 7 geriatrics-05-00026-f007:**
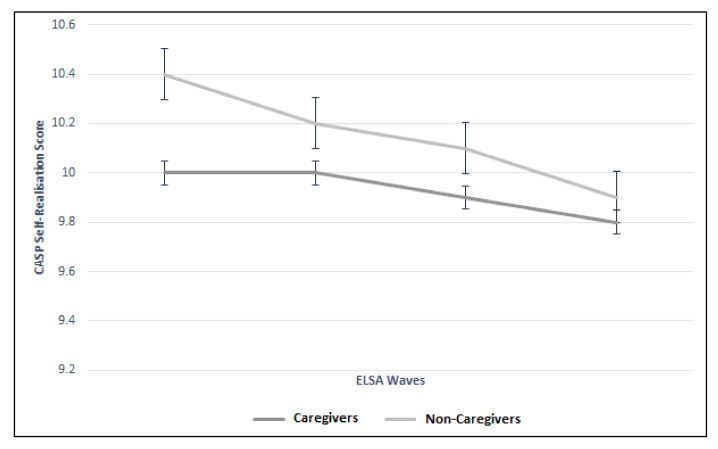
Trajectory of self-realization CASP score between caregiver and non-caregiver cohorts across four ELSA waves.

**Table 1 geriatrics-05-00026-t001:** Summary of demographic and biopsychosocial factors characterizing the informal caregiver (cases) and non-informal caregiver (controls).

	Caregivers (Cases; *N* = 1375)	Non-Caregiver (Controls; *N* = 2750)	Univariate Analysis (*p*-value; 95% CI—confidence interval)
Age (mean; SD)	62.0 (9.9)	61.5 (9.5)	0.13 (−0.14 to 1.01)
Ethnic Group (Caucasian; %)	603 (98.0)	2454 (97.6)	0.00 (−1.32 to −1.15)
Gender (female; %) (N = 11,730)	865 (62.9)	1705 (62.0)	<0.01 (0.09 to 0.353
Relationship (*n*; %)			
Married	1087 (79.1)	1866 (67.9)	<0.01 (−0.38 to −0.22)
Cohabit	58 (4.2)	133 (4.8)	
Neither	230 (16.7)	751 (27.3)	
Employment status			
Retired	594 (43.2)	1125 (41.0)	0.48 (−0.01 to 0.00)
Employed	382 (27.8)	966 (35.1)	
Self-employed	69 (5.0)	200 (7.3)	
Unemployed	20 (1.5)	24 (0.9)	
Permanently sick/disabled	73 (5.3)	150 (5.5)	
Looking after home or family	225 (16.4)	245 (8.9)	
Not reported	12 (0.9)	40 (1.5)	
Relationship to care-recipient (*n*; %)			
Spouse	615 (44.7)		
Parent	378 (27.5)		
Parent in law	93 (6.8)		
Other relative	128 (9.3)		
Friend or neighbor	167 (12.1)		
Not reported			
Hours caregiving in past week (mean; SD; *n* = 376)	56.8 (70.2)		
Number of members in household (mean; SD)	2.3 (0.9)	2.1 (0.9)	<0.01 (0.07 to 0.19)
Self-reported			
Excellent	73 (5.3)	192 (7.0)	0.03 (−0.05 to 0.01)
Very good	215 (15.6)	446 (16.2)	
Good	244 (17.8)	456 (16.6)	
Fair	117 (8.5)	250 (9.1)	
Poor	25 (1.8)	69 (2.5)	
Not reported	701 (51.0)	1337 (48.6)	
Self-reported chronic diseases (yes; %)	766 (55.7)	1457 (53.0)	0.10 (−0.24 to 0.02)
Often troubled by pain (yes; %)	583 (42.4)	1027 (37.4)	<0.01 (−0.34 to −0.08)
Immediate word recall (mean; SD)	5.75 (1.62)	5.45 (1.78)	0.05 (−0.02 to −0.00)
Delayed word recall (mean; SD)	4.36 (2.00)	4.07 (2.08)	0.82 (−0.01 to 0.01)
Fluency score (mean; SD)	20.34 (6.10)	19.30 (6.13)	0.02 (−0.01 to −0.00)
Numeracy score (mean; SD)	4.05 (1.24)	4.05 (1.30)	0.02 (0.00 to 0.03)
Prospective memory score (mean; SD)	5.57 (2.40)	5.35 (2.54)	0.61 (−0.01 to 0.00)
Self-reported depression (yes; %)	259 (18.8)	449 (16.3)	0.13 (−0.29 to 0.04)
Self-reported loneliness (yes; %)	141 (10.3)	323 (11.8)	0.06 (−0.01 to 0.39)
Sociological Measures			
Frequency went to cinema			
Twice a month or more	22 (1.6)	50 (1.8)	<0.01 (0.03 to 0.13)
About once a month	62 (4.5)	113 (4.1)	
Every few months	140 (10.2)	363 (13.2)	
Once or twice a year	204 (14.8)	472 (17.2)	
Less than once a year	221 (16.1)	455 (16.6)	
Never	726 (52.8)	1297 (47.2)	
Frequency ate out			
Twice a month or more	514 (37.4)	1114 (40.5)	<0.01 (0.04 to 0.13)
About once a month	272 (20.0)	594 (21.6)	
Every few months	262 (19.1)	509 (18.5)	
Once or twice a year	160 (11.6)	299 (10.9)	
Less than once a year	35 (2.6)	62 (2.3)	
Never	132 (9.6)	172 (6.3)	
Frequency visited art gallery/museum			
Twice a month or more	22 (1.6)	58 (2.1)	0.03 (0.01 to 0.11)
About once a month	58 (4.2)	96 (3.5)	
Every few months	159 (11.6)	352 (12.8)	
Once or twice a year	296 (21.5)	636 (23.1)	
Less than once a year	190 (13.8)	468 (17.0)	
Never	650 (47.3)	1140 (41.5)	
Frequency visited theatre, concert, opera			
Twice a month or more	24 (1.8)	52 (1.9)	0.01 (0.02 to 0.11)
About once a month	73 (5.3)	148 (5.4)	
Every few months	239 (17.4)	515 (18.7)	
Once or twice a year	300 (21.8)	694 (25.2)	
Less than once a year	178 (13.0)	360 (13.1)	
Never	561 (40.8)	981 (35.7)	
Holiday in UK in last 12 months (yes; %)	811 (59.0)	1628 (59.2)	0.73 (−0.10 to 0.15)
Holiday abroad in last 12 months (yes; %; *N* = 10,755)	599 (43.6)	1454 (52.9)	<0.01 (−0.44 to −0.18)
Daytrips last 12 months (yes; %)	940 (68.4)	2007 (73.0)	0.02 (−0.29 to −0.03)
Use the internet/email (yes; %)	434 (31.6)	993 (36.1)	0.02 (−0.28 to −0.02)
Attend political party, trade union, environmental group (yes; %)	181 (13.2)	408 (14.8)	0.21 (-0.05 to 0.20)
Member of residential group (yes; %)	262 (19.1)	475 (17.3)	<0.01 (0.06 to 0.33)
Church or religious member (yes; %)	326 (23.7)	531 (19.3)	<0.01 (0.15 to 0.44)
Member of charitable organization (yes; %)	297 (21.6)	460 (16.7)	<0.01 (0.19 to 0.50)
Attends education, arts, music group (yes; %)	192 (14.0)	368 (13.4)	0.01 (0.03 to 0.31)
Attend social club (yes; %)	240 (17.5)	559 (20.3)	0.55 (−0.08 to 0.14)
Attend sports club, gym or evening class (yes; %)	263 (19.1)	571 (20.8)	0.23 (−0.04 to 0.18)
Attends another organization or club (yes; %)	294 (21.4)	691 (25.1)	0.88 (−0.09 to 0.11)
Psychological Measures			
GHQ-12 (General Health Questionnaire-12): concertation (*n*; %)			
Better than usual	32 (2.3)	71 (2.6)	<0.01 (0.07 to 0.34)
Same as usual	1167 (84.9)	2374 (86.3)	
Less than usual	154 (11.2)	246 (9.0)	
Much less than usual	22 (1.6)	43 (1.6)	
GHQ-12: loss sleep due to worry (*n*; %)			
Better than usual	437 (31.8)	1032 (37.5)	<0.01 (0.18 to 0.35)
Same as usual	691 (50.3)	1372 (49.9)	
Less than usual	186 (13.5)	268 (9.8)	
Much less than usual	61 (4.4)	62 (2.3)	
GHQ-12: perceived value (*n*; %)			
Better than usual	120 (8.7)	194 (7.1)	0.68 (−0.08 to 0.12)
Same as usual	1112 (80.9)	2247 (81.7)	
Less than usual	111 (8.1)	209 (7.6)	
Much less than usual	32 (2.3)	82 (3.0)	
GHQ-12: capable of decision-making (*n*; %)			
Better than usual	76 (5.5)	164 (6.0)	0.14 (−0.04 to 0.25)
Same as usual	1224 (86.0)	2408 (87.6)	
Less than usual	68 (5.0)	142 (5.2)	
Much less than usual	7 (0.5)	20 (0.7)	
GHQ-12: constantly under strain (*n*; %)			
Better than usual	282 (20.5)	754 (27.4)	<0.01 (0.24 to 0.42)
Same as usual	754 (54.8)	1550 (56.4)	
Less than usual	282 (20.5)	366 (13.3)	
Much less than usual	56 (4.1)	63 (2.3)	
GHQ-12: unable to overcome difficulties (*n*; %)			
Better than usual	420 (30.6)	1052 (38.3)	<0.01 (0.15 to 0.33)
Same as usual	779 (56.7)	1411 (51.3)	
Less than usual	146 (10.6)	205 (7.5)	
Much less than usual	30 (2.2)	65 (2.4)	
GHQ-12: able to enjoy life (*n*; %)			
Better than usual	65 (4.7)	128 (4.7)	0.01 (0.05 to 0.28)
Same as usual	1099 (79.9)	2261 (82.2)	
Less than usual	177 (12.9)	279 (10.2)	
Much less than usual	34 (2.5)	66 (2.4)	
GHQ-12: resilience (*n*; %)			
Better than usual	64 (4.7)	119 (4.3)	<0.01 (0.15 to 0.33)
Same as usual	1181 (85.9)	2405 (87.5)	
Less than usual	109 (7.9)	170 (6.2)	
Much less than usual	21 (1.5)	40 (1.5)	
GHQ-12: unhappy and depressed (*n*; %)			
Better than usual	557 (40.5)	1268 (46.1)	<0.01 (00.01 to 0.25)
Same as usual	483 (42.4)	1080 (39.3)	
Less than usual	195 (14.2)	314 (11.4)	
Much less than usual	40 (2.9)	72 (2.6)	
GHQ-12: losing confidence in self (*n*; %)			
Better than usual	660 (48.0)	1400 (50.9)	<0.01 (0.04 to 0.20)
Same as usual	543 (39.5)	1034 (37.6)	
Less than usual	140 (10.2)	245 (8.9)	
Much less than usual	32 (2.3)	54 (2.0)	
GHQ-12: perceived worth (*n*; %)			
Better than usual	987 (71.8)	1982 (72.1)	0.14 (−0.02 to 0.16)
Same as usual	309 (22.5)	601 (21.9)	
Less than usual	63 (4.6)	113 (4.1)	
Much less than usual	16 (1.2)	37 (1.4)	
GHQ-12: perceived happiness (*n*; %)			
Better than usual	127 (9.2)	261 (9.5)	0.01 (0.04 to 0.29)
Same as usual	1122 (81.6)	2270 (82.6)	
Less than usual	105 (7.6)	166 (6.0)	
Much less than usual	21 (1.5)	36 (1.3)	

**Table 2 geriatrics-05-00026-t002:** Summary of the cross-sectional caregiver versus non-caregiver multivariate analysis results.

Variable	Odd Ratios	95% Confidence Intervals	*p*-Value
**Model 1 (Biological–physical health factors)**			
Ethnicity	1.24	1.15 to 1.32	<0.01
Gender	0.23	0.07 to 0.39	0.01
Self-rated health	0.01	−0.05 to 0.26	0.54
Self-rated chronic diseases	0.13	−0.30 to 0.03	0.11
Often troubled by pain	0.17	0.00 to 0.33	0.05
Immediate word recall	0.03	−0.02 to 0.08	0.28
Fluency score	0.01	−0.01 to 0.02	0.31
Numeracy score	0.03	−0.10 to 0.03	0.33
Self-reported loneliness	0.35	0.11 to 0.56	<0.01
**Model 2 (Biopsychological factors)**			
Ethnicity	1.29	1.20 to 1.37	<0.01
Gender	0.27	0.10 to 0.43	<0.01
Often troubled by pain	0.08	−0.24 to 0.08	0.33
Self-reported loneliness	0.24	−0.00 to 0.49	0.05
Relationship status	0.32	0.21 to 0.44	<0.01
Cinema attendance	0.04	−0.03 to 0.11	0.24
Eats out	0.05	−0.01 to 0.11	0.09
Art gallery	0.05	−0.02 to 0.12	0.20
Theatre	0.05	−0.02 to 0.13	0.13
Holiday abroad	0.36	0.19 to 0.53	<0.01
Day trip	0.12	−0.31 to 0.06	0.20
Internet	−0.4	−0.32 to 0.04	0.14
Number in household	0.07	−0.03 to 0.17	0.17
Residential group	0.12	−0.08 to 0.32	0.25
Church	0.32	0.12 to 0.52	<0.01
Charity member	0.47	0.26 to 0.68	<0.01
Education class	0.27	0.03 to 0.51	0.03
**Model 3 (Biopsychosocial factors)**			
Ethnicity	1.28	1.20 to 1.37	<0.01
Gender	0.25	0.09 to 0.41	<0.01
Self-reported loneliness	0.26	0.01 to 0.51	0.05
Relationship status	0.36	0.16 to 0.46	<0.01
Holiday abroad	0.51	0.35 to 0.66	<0.01
Church	0.30	0.11 to 0.49	<0.01
Charity member	0.35	0.14 to 0.55	<0.01
Education class	0.11	−0.11 to 0.33	0.33
Concentration	0.10	−0.11 to 0.31	0.33
Sleep	0.05	−0.08 to 0.18	0.46
Strain	0.23	0.09 to 0.37	<0.01
Problem-solving	0.05	−0.09 to 0.20	0.48
Enjoyment	0.07	−0.27 to 0.13	0.49
Resilience	0.06	−0.17 to 0.30	0.58
Depression	0.04	−0.11 to 0.19	0.61
Confidence	0.06	−0.20 to 0.08	0.39
Happiness	0.05	−0.13 to 0.24	0.58

**Table 3 geriatrics-05-00026-t003:** Summary of the trajectories of psychosocial variables for caregivers and non-caregivers across five English Longitudinal Study of Ageing (ELSA) Waves (10 years; caregivers: 777 vs. 1463 non-caregivers).

Variable	Group	Wave 1	Wave 2	Wave 3	Wave 4	Wave 5	*p*-value (95% CI)
Self-reported loneliness (yes; %)	Caregiver	70 (9.0)	92 (11.9)	92 (11.9)	103 (13.3)	105 (13.5)	<0.01 (0.05 to 0.22)
	Non-Caregiver	131 (9.1)	149 (10.4)	146 (10.2)	148 (10.3)	166 (11.6)	
Relationship status (married/cohabiting; %)	Caregiver	639 (82.2)	587 (75.6)	562 (72.8)	547 (70.4)	515 (66.4)	0.01 (−0.15 to −0.02)
	Non-Caregiver	1070 (74.5)	1062 (74.0)	1044 (72.8)	1036 (72.2)	1101 (70.4)	
Holiday abroad (yes; %)	Caregiver	371 (47.8)	371 (47.8)	346 (44.7)	332 (42.7)	319 (41.1)	0.34 (−0.08 to 0.03)
	Non-Caregiver	803 (55.9)	753 (52.4)	718 (50.0)	694 (48.4)	640 (44.6)	
Church membership (yes; %)	Caregiver	198 (25.5)	196 (25.2)	191 (24.7)	183 (23.6)	184 (23.7)	0.49 (−0.15 to 0.07)
	Non-Caregiver	319 (22.2)	288 (20.0)	283 (19.7)	268 (18.7)	266 (18.5)	
Charity group membership (yes; %)	Caregiver	178 (22.9)	160 (20.6)	170 (21.9)	153 (20.0)	156 (20.1)	0.87 (−0.12 to 0.10)
	Non-Caregiver	272 (18.9)	244 (17.0)	242 (16.9)	242 (16.9)	273 (19.0)	
Total CASP score (mean; SD)	Caregiver	No Data	42.4 (8.8)	41.2 (8.4)	40.9 (8.7)	41.1 (8.6)	0.91 (−0.03 to 0.03)
	Non-Caregiver	No Data	43.5 (8.2)	42.2 (8.3)	41.9 (8.4)	41.4 (8.9)	
Control CASP (mean; SD)	Caregiver	No Data	8.2 (2.6)	7.6 (2.5)	7.6 (2.5)	7.6 (2.4)	<0.01 (0.03 to 0.12)
	Non-Caregiver	No Data	8.5 (2.5)	8.0 (2.4)	7.8 (2.4)	7.7 (2.5)	
Autonomy CASP (mean; SD)	Caregiver	No Data	10.7 (2.8)	10.1 (2.7)	10.1 (2.8)	10.2 (2.6)	<0.01 (0.02 to 0.10)
	Non-Caregiver	No Data	10.9 (2.6)	10.6 (2.6)	10.6 (2.7)	10.5 (2.7)	
Pleasure CASP (mean; SD)	Caregiver	No Data	13.3 (2.3)	13.3 (2.1)	13.3 (2.2)	13.3 (2.2)	0.01 (−0.12 to −0.02)
	Non-Caregiver	No Data	13.5 (2.2)	13.3 (2.1)	13.2 (2.2)	13.2 (2.3)	
Self-realization CASP (mean; SD)	Caregiver	No Data	10.0 (3.3)	10.0 (2.9)	9.9 (3.1)	9.8 (3.2)	Omitted to collinearity
	Non-Caregiver	No Data	10.4 (3.1)	10.2 (3.0)	10.1 (3.1)	9.9 (3.2)	

**Table 4 geriatrics-05-00026-t004:** Presentation of basic demographic characteristics for cross-section cohort and trajectory cohort characteristics for caregiver and non-caregiver cases.

	Caregivers (Cases)			Non-Caregivers (Controls)		
	Cross-sectional Cohort *N* = 1375	Trajectory Cohort *N* = 777	*p*-value	Cross-Sectional Cohort *n* = 2750	Trajectory Cohort *n* = 1463	*p*-value
Age (mean; SD)	62.0 (9.9)	61.7 (9.8)	0.361	61.5 (9.5)	60.9 (9.6)	0.492
Ethnic Group (Caucasian; %)	603 (98.0)	758 (97.6)	0.791	2454 (97.6)	1430 (97.8)	0.735
Gender (female; %)	865 (62.9)	479 (61.7)	0.673	1705 (62.0)	901 (61.6)	0.302
Relationship (*n*; %)						
Married	1087 (79.1)	608 (78.3)	0.365	1866 (67.9)	973 (66.5)	0.581
Cohabit	58 (4.2)	32 (4.1)		133 (4.8)	73 (5.0)	
Neither	230 (16.7)	137 (17.6)		751 (27.3)	417 (28.5)	
Hours caregiving in past week (mean; SD)	56.8 (70.2)	54.7 (71.2)	0.164	54.3 (69.3)	52.3 (70.6)	0.236

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
