# Peer review of "Trajectory of Psychosocial Measures Amongst Informal Caregivers: Case-Controlled Study of 1375 Informal Caregivers from the English Longitudinal Study of Ageing"

_geriatrics, 2020, doi:10.3390/geriatrics5020026_

Round 1

Reviewer 1 Report

Thank you for the opportunity to review this this interesting paper. The study is well written, but the introduction is very short. I suggest to describe more exstensively the main results of the studies reported.

Author Response

Comment: Thank you for the opportunity to review this this interesting paper. The study is well written, but the introduction is very short. I suggest to describe more extensively the main results of the studies reported.

Response: Thank you for this kind comment. As suggested, we have revised the Introduction to provide more context to the research question and overarching study aim (Introduction, Paragraph 1, Lines 48-51 & 57 to 58; Paragraph 2, Lines 61-67; Paragraph 4, Lines 83-87).

Reviewer 2 Report

This is an interesting study that provides a unique longitudinal evaluation of caregiver burden in a nationally-represented sample. The approach is theory-based and the selection of Engel's biopsychosocial model to frame the investigation is appropriate. The data are limited by the scope of the parent study (ELSA) but the sample size of caregivers and non-caregivers is large which improves the validity and generalizability of the results. Scientific rigor is demonstrated in the application of the STROBE guidelines, confirmation of the caregiver role at each data collection wave, and inclusion of an appropriate control/comparator group.

Introduction: Because the cohort are aged 50 or older, more emphasis could be placed on older caregivers of adult patients and the literature on older caregivers or whether the literature does not adequately address the burdens on older informal caregivers. The purpose statement focuses on the psychosocial features of caregivers that could guide the forms of support that could benefit them; 

Methods: if available, describe the proportion of respondents from urban versus rural communities and consider this in the data analysis plan. The date span of data collection is not clear; although this information may be available in the seminal paper about ELSA, it would be helpful to readers to know how current the data are given changes in society and medical care. The selection of outcome variables is appropriate and matches biopsychosocial theory. It would be helpful to provide the range of scores for the CASP outcomes. The statistical analyses are appropriate and the number of cases in the trajectory analysis was provided.

Results: were the cases in the trajectory analysis different in any pertinent ways from the cases included at baseline (Table 1)? That is, is there measurable evidence of selection bias in the trajectory analysis?  Did the retention rate differ between caregivers and non-caregivers? The figures for the changes in CASP outcomes have differ in the range of the Y-axis, making the results difficult to interpret. As mentioned above, the range of CASP outcomes could be provided in the methods and/or in the titles of the figures. Although the difference in "often troubled by pain" was marginally significant (P=0.05), consider expanding on the interpretation of this biological outcome and how pain might interfere with caregivers' participation in travel and social events.

Are figures 4 and 5 across 4 ELSA waves and not 5 waves as stated in the titles?

Discussion: Another strength of the study is that the caregivers were not selected based on the illness/condition of the patient, other than the patient being an adult. It's worth stating (and citing) in the discussion that studies of informal caregiver burden over time have been published but are disease/condition specific.

The strengths and weaknesses of the study are acknowledged.

Table 3: in the Wave 1 column, change "not data" to "no data" or N/A (not applicable).

Author Response

Reviewer 2

Comment: This is an interesting study that provides a unique longitudinal evaluation of caregiver burden in a nationally-represented sample. The approach is theory-based and the selection of Engel's biopsychosocial model to frame the investigation is appropriate. The data are limited by the scope of the parent study (ELSA) but the sample size of caregivers and non-caregivers is large which improves the validity and generalizability of the results. Scientific rigor is demonstrated in the application of the STROBE guidelines, confirmation of the caregiver role at each data collection wave, and inclusion of an appropriate control/comparator group.

Response: Thank you for your kind words. No amendment required in response to this comment.

Comment: Introduction: Because the cohort are aged 50 or older, more emphasis could be placed on older caregivers of adult patients and the literature on older caregivers or whether the literature does not adequately address the burdens on older informal caregivers. The purpose statement focuses on the psychosocial features of caregivers that could guide the forms of support that could benefit them; 

Response: As suggested, we have provided further evidence to provide the context around older people who are caregiver as we agree this needs emphasising to set-the-scene of this paper (Introduction, Paragraph 2, Lines 67-71).

Comment: Methods: if available, describe the proportion of respondents from urban versus rural communities and consider this in the data analysis plan.

Response: Unfortunately this data were not gathered from the cohort and not identifiable within the ELSA cohort we analysed. This has been highlighted as a limitation (Discussion, Paragraph 6, Lines 314-316).

Comment: Methods: The date span of data collection is not clear; although this information may be available in the seminal paper about ELSA, it would be helpful to readers to know how current the data are given changes in society and medical care.

Response: We have provided the date span for Waves 1 to 5 as suggested (Methods, Participant Identification, Paragraph 1, Line 115).

Comment: Methods: The selection of outcome variables is appropriate and matches biopsychosocial theory.

Response: Thank you. No amendment required to this comment.

Comment: Methods: It would be helpful to provide the range of scores for the CASP outcomes.

Response: We have revised the text to provide range scores for both the GHQ-12 and CASP-19 (for completeness)(Methods, Data Identification, Paragraph 2, Lines 127-128).

Comment: Methods: The statistical analyses are appropriate and the number of cases in the trajectory analysis was provided.

Response: Thank you. No amendment required to this comment.

Comment: Results: were the cases in the trajectory analysis different in any pertinent ways from the cases included at baseline (Table 1)? That is, is there measurable evidence of selection bias in the trajectory analysis? 

Response: The was limited evidence for selection bias between the caregiver and non-caregiver when we assessed basic demographic characteristics for the trajectory and cross-sectional cohorts. This data and interpretation for these analyses are presented in the text (Results, Trajectory Analysis, Lines 226 to 228) and (Table 4).

Comment: Results: Did the retention rate differ between caregivers and non-caregivers? The figures for the changes in CASP outcomes have differ in the range of the Y-axis, making the results difficult to interpret.

Response: Participants were included in the analysis if they provided with full-data across the waves. Retention rate across the waves was therefore full for both caregiver and non-caregiver cohorts. This has been clarified in the text (Methods, Participant Identification, Paragraph 2, Lines 2-3) and is presented in Figure 1 study flow-chart.

Comment: As mentioned above, the range of CASP outcomes could be provided in the methods and/or in the titles of the figures.

Response: As suggested, we have revised the text to provide range scores for both the GHQ-12 and CASP-19 (for completeness)(Methods, Data Identification, Paragraph 2, Lines 127-128).

Comment: Results: Although the difference in "often troubled by pain" was marginally significant (P=0.05), consider expanding on the interpretation of this biological outcome and how pain might interfere with caregivers' participation in travel and social events.

Response: Thank you. As ‘often troubled by pain’ dropped out of multivariate analysis at Model 2 (p=0.33), we feel uncomfortable about emphasising this findings further, given that it did not remain independently associated in the latter models. This suggest as implied by the reviewer, participation in travel and social events may have greater independence in this analysis over pain. Accordingly we have elected not to explore pain further in this paper, to avoid diluting the message and confusing the reader. If the reviewer feels strongly about this, we would be happy to reconsider this decision.

Comment: Are figures 4 and 5 across 4 ELSA waves and not 5 waves as stated in the titles?

Response: Thank you for correcting this. We have revised the tables as suggested (Figure 4-7).

Comment: Discussion: Another strength of the study is that the caregivers were not selected based on the illness/condition of the patient, other than the patient being an adult. It's worth stating (and citing) in the discussion that studies of informal caregiver burden over time have been published but are disease/condition specific.

Response: Thank you for suggesting this point. As suggested, we have incorporated this into the Discussion as a strength of the study (Discussion, Paragraph 6, Lines 343-346).

Comment: The strengths and weaknesses of the study are acknowledged.

Response: Thank you. No amendment required to this comment.

Comment: Table 3: in the Wave 1 column, change "not data" to "no data" or N/A (not applicable).

Response: Thank you for highlighting this. We have made this correction (Table 3, Column 3(Wave 1), Rows 12-21).

Reviewer 3 Report

This is an interesting paper on an important subject. Since most studies in the research area use (both qualitative and quantitative) cross-sectional design, this paper add to the body of knowlege with its longitudinal design, additionally using, nationally representative data - from the well-known ELSA cohort. I have one concern. The authors mention, very briefly, different theoretical models in the Introduction and follow them up under Discussion. From my point view,  these are grounded in different perspectives - for example the stess-model and the Life course model. I think it will strenghten the paper to use, one or two of these models. If not elaborated on more (I suppose this is not possible regarding the word limit), it confuses more than it add.

Author Response

Reviewer 3

Comment: This is an interesting paper on an important subject. Since most studies in the research area use (both qualitative and quantitative) cross-sectional design, this paper add to the body of knowledge with its longitudinal design, additionally using, nationally representative data - from the well-known ELSA cohort. I have one concern.

Response: Thank you for these comments. We have addressed the comment raised below.

Comment: The authors mention, very briefly, different theoretical models in the Introduction and follow them up under Discussion. From my point view,  these are grounded in different perspectives - for example the stress-model and the Life course model. I think it will strengthen the paper to use, one or two of these models. If not elaborated on more (I suppose this is not possible regarding the word limit), it confuses more than it add.

Response: Thank you for this comment. We have reflected on this comment. Whilst we understand how this may aid the interpretation of the paper, to maintain the paper’s clarity to message, we have elected to maintain the text on theoretical models in the Introduction and Discussion, but have not elaborate further on these, with the concern that, as suggested by the reviewer, that this may confuse the message. If the editorial team and reviewer feel further reflection on this is needed, we would be happy to reconsider this current decision.

Round 2

Reviewer 1 Report

I agree with your revision.

Author Response

Thank you.